# CodeIt: Abstract Reasoning with Iterative Policy-Guided Program Synthesis

## Abstract

Artificial intelligence systems are increasingly solving tasks that are commonly believed to require human-like reasoning ability. However, learned approaches still fare poorly on the Abstraction and Reasoning Corpus (ARC), a benchmark that measures skill-acquisition efficiency as a proxy for intelligence. Each ARC task requires an agent to reason about a transformation between input and output pairs. In this work, we solve these tasks by identifying the program that applies this transformation. We propose CodeIt, a program synthesis approach that leverages a higher level of abstraction through a domain-specific language. CodeIt iterates between sampling from the current large language model policy and learning that policy using supervised learning. The sampling stage augments newfound programs using hindsight relabeling and program mutation, requiring no expert search procedure. We demonstrate CodeIt's effectiveness on the ARC benchmark, where we show that learning to write code in iterations leads to inter-task generalization, which results in state-of-the-art performance.

## 1 Introduction

Iterative learning methods such as Expert Iteration (Anthony et al., 2017) have achieved super-human performance in games such as chess, go (Silver et al., 2016; 2018), hex (Anthony et al., 2017), and combinatorial problems such as bin-packing (Laterre et al., 2019). In these tasks, humans typically need to reason and think multiple steps ahead to select an optimal next move. By playing through many games and looking back at what worked well and what did not, experts acquire intuition for selecting good moves. Iterative learning methods emulate this process by alternating between two phases: gathering data with an exploration policy, and improving the policy by training on the newfound experiences. This process works well in narrow domains, but has yet to show results in domains that require generalization between different tasks.

We broaden the application scope of iterative learning methods with the Abstraction and Reasoning Corpus (ARC) (Chollet, 2019), a benchmark dataset for intelligence with a focus on generalization ability. It aims to measure *skill-acquisition efficiency over a range of different tasks*, argued by Chollet (2019) to be a proxy metric for intelligence. Intuitively, if two agents gather a similar amount of experience in a previously unknown set of tasks, the one that performs better has acquired the necessary skills more efficiently, and can be said to be more intelligent. The ARC dataset consists of 400 training tasks and 400 evaluation tasks, plus 200 hidden test tasks not available to the public and meant to score competitors to the ARC challenge.[1] Each task contains one or more demonstration examples, and one or more test examples, where each example is an input-output pair of *grids*. We show a toy example of an ARC task in Figure 1. Based on the demonstration examples, an agent needs to reason about what output grid the test inputs should map to.

Seen through the lens of program synthesis, ARC is an instance of programming-by-examples with grids as inputs and outputs. The programming language to be used is not specified, which makes it an *open-DSL* problem: one is free to define a custom domain-specific language, and therefore, a custom search space. The limited size of the training dataset and the difficulty of individual problems make it essential for the agent to generalize between different tasks. Existing approaches either fail to generalize (Ainooson et al., 2023; Mirchandani et al., 2023) or are too inefficient to apply on the full dataset (Alford et al., 2021; Kolev et al., 2020; Xu et al., 2022; Park et al., 2023).

---

[1] https://lab42.global/arcathon/

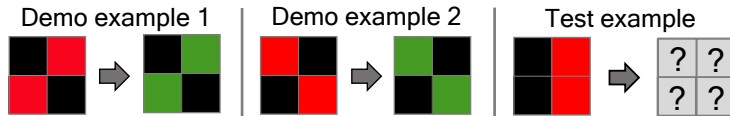

Figure 1: A simplified ARC task. Given two demonstration examples, the goal is to determine the output grid for the test example in three attempts or fewer. The size of the grids and the number of demonstration and test examples may be different for each task.

In this work, we propose a scalable program synthesis approach that benefits from inter-task generalization. The program synthesis approach induces a bias toward generalizable solutions through a domain specific programming language. We empirically validate our method on the ARC dataset, where the goal for each task then becomes to write the *program* that, when applied on the input grids, produces the target output grids. Our approach, which we call Code Iteration or *CodeIt* for short, iterates between a sampling stage and a learning stage. In the sampling stage, we sample new programs from a language model policy, conditioned on the input-output examples of ARC tasks. We then execute these programs on their associated inputs, and add the resulting input-output pairs and corresponding programs to a replay buffer. In the learning stage, we train the policy on experiences sampled from the replay buffer. The updated policy is then used in the next iteration of sampling. This iterative procedure thus allows us to automatically generate new data without human intervention.

CodeIt solves 48/400 ARC evaluation tasks, achieving state-of-the-art performance in line with GPT-4 (Gendron et al., 2023). It achieves this result by gaining useful experience over the full dataset due to to its scalability, and by successfully generalizing this experience between different tasks.

## 2 METHOD

We approach the ARC problem as a programming-by-examples problem: for a given set of tasks that we call the *search set*, we aim to find programs that correctly match inputs with their respective outputs, and we do so by training a *policy* to produce programs when shown demonstration examples. This is achieved by pretraining the policy on ground truth data, and then iterating between two stages: writing programs using a policy, and learning from the program outputs. We first describe key design choices below, and then explain the iterative procedure.

### 2.1 DESIGN CHOICES

**Programming language**  We restrict our programming language to the open source domain specific language (DSL) of Hodel (2023), designed specifically for the ARC training split. This DSL contains grid manipulation functions (e.g., `vmirror` or `hmirror`, which mirror the grid along the vertical or horizontal axis), `fill` functions that replace all pixels of a certain color, and functions that return locations of specific pixel groups. See Appendix B.4 for details on the DSL and more example primitives, and see (Hodel, 2023) for discussion on the DSL's primitives and capability.

**Policy model**  Our choice of policy is a pretrained encoder-decoder Large Language Model (LLM). We use the 220 million parameter CodeT5+ (Wang et al., 2023b) model and its default tokenizer, which are pretrained on a diverse set of programming tasks. We input the demonstration examples to the encoder, and let the decoder generate the corresponding program. If necessary, demonstration examples are truncated to fit in the encoder context window.

**Grid representation**  In order to condition the language model policy on input-output grids, we represent them as text. Instead of encoding the grid as a 2-dimensional array, we use an object-centric text representation. Specifically, each color is encoded as an integer, and for each color in the grid we list all the grid cells with that color as $[x, y]$ coordinates. Since the majority of cells belong to the background color, this procedure significantly reduces the number of tokens required to encode the grid (see Figure 5 in the Appendix). As an example, the demonstration examples from Figure 1 would be represented as follows:

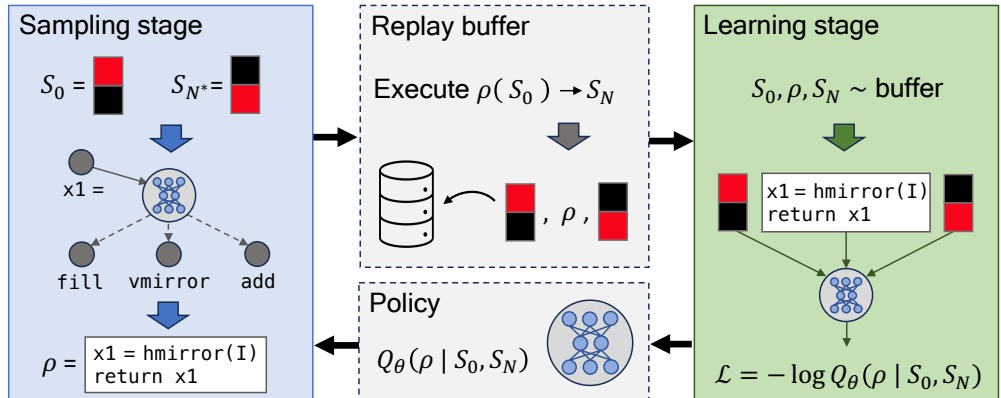

Figure 2: Overview of the iterative algorithm. The sampling stage produces new programs $\rho$. These are run on inputs $S_0$, and the resulting input-output-program triplets are stored in the replay buffer. The learning stage uses samples from the buffer to finetune the policy network. The updated policy is then used in the next sampling stage.

```
2x2  bg=0  1=0,1  1,0|2x2  bg=0  1=1,0  0,1
2x2  bg=0  2=0,1  1,0|2x2  bg=0  2=1,0  0,1<eos>
```

This object-centric text representation, similar to the one of Xu et al. (2023), works well for sparse grids and is human-interpretable.

## 2.2 RUNNING CODE ITERATION

We always initialize the approach from ground truth data, and use this data to pre-train a policy model, and to initialize a replay buffer. We then start the Code Iteration procedure, iterating between a *sampling* and *learning* stage. We refer to one full pass of sampling and learning as a *meta-iteration*. We show the procedure in Fig. 2, and explain each stage in more detail below. For pseudocode, see Appendix A.1.

**Initialization**  We start from a dataset of ARC training tasks, and solution programs written in the domain-specific language (DSL) of Hodel (2023). This dataset is expanded by randomly mutating programs (for details of this procedure, see Appendix A.2). We pre-train the policy network on the resulting set, and initialize our replay buffer with it.

This dataset augmentation step serves multiple purposes. Pretraining teaches the model the DSL syntax, and enables the model to learn how to interpret the task demonstration examples before we begin iterative learning, improving the quality of our policy samples in early meta-iterations. Mixing in mutated programs also acts as a form of regularisation, and is a common approach in iterative policy improvement for program synthesis (Ellis et al., 2020; Fawzi et al., 2022).

**Sampling stage**  In the sampling stage, we sample new programs using the policy $Q_\theta$. Let the *search set* be the set of tasks for which we want to find a corresponding program. For each task in the search set, we convert the demonstration examples' input $S_0$ and target output $S_{N*}$ from grid to text representation, encode these using the policy, and then autoregressively decode a program: $\rho \sim Q_\theta(.|S_0, S_{N*})$. We then run the obtained program on the input grids. If the program is syntactically incorrect or the runtime is too high, we discard it. Otherwise, we obtain a program output $S_N = \rho(S_0)$, and can add a new triplet to the buffer, consisting of the demonstration inputs $S_0$, the program $\rho$, and the obtained outputs $S_N$ (which may or may not match the target $S_{N*}$). In each sampling stage we repeat this procedure $n_\rho$ times per task, where $n_\rho$ is a hyperparameter.

Replacing the target output by the realized one is similar to hindsight experience replay (Andrychowicz et al., 2017), and ensures that we obtain an experience every time we find a syntactically correct program, thereby preventing stagnation of the buffer. Note that although these programs may not solve the tasks we are interested in, they are always valid in terms of syntax and semantics (correctly

mapping $S_0$ to $S_N$). They can therefore be used to teach the policy about program syntax and program behaviour, which may lead to positive transfer to the search set. We emphasize that we never add test examples nor performance on the test examples to our buffer, as one should not have access to their target output grid during sampling.

**Learning stage**   During the learning stage, the policy $Q_\theta$ is trained on experiences from the buffer, consisting of input grids $S_0$, the program $\rho$, and corresponding output grids $S_N$. The training objective is then a straightforward negative log-likelihood objective:

$$\mathcal{L}(S_0, \rho, S_N) = -\log Q_\theta(\rho|S_0, S_N). \tag{1}$$

We keep only a single copy of the policy network which we continue to update during each learning stage. In particular, we do not compare with past versions to guarantee an improvement in the policy before using it in the next sampling stage. This could lead to worse performance in terms of finding the correct program in the next iteration, but in practice we find this is not a problem.

To bias our training samples towards high quality experiences, we sample experiences that solve ARC tasks more often than those that solve artificial tasks. This also prevents forgetting programs that solve tasks of interest.

## 3   EXPERIMENTS

In this section, we aim to verify the efficacy of CodeIt. We first tuned the hyperparameters of CodeIt on a custom training and validation split (for details, see Appendix B). Using these hyperparameters, we benchmark our method on the ARC evaluation split and compare against previous state-of-the-art methods. Finally, we ablate the importance of individual components of CodeIt.

We define *demonstration performance* as the percentage of solved demonstration examples on a given task. To evaluate if set of test examples for a given task are solved, we sort our solutions first by task demonstration performance and then by program length, and evaluate the top three programs on the set of test examples. Following ARC evaluation procedure, if at least one of these three programs maps all test example inputs to outputs, the task is solved and *test performance* is 1. We emphasize that the search procedure only makes use of the demonstration performance, and that we use the test performance solely for final evaluation.

### 3.1   BASELINES

**Mutating programs with task relabelling**   The mutation baseline keeps mutating the set of training programs provided by Hodel (2023), and is essentially a random search that stays close to a set of ground truth programs. Specifically, for each meta-iteration, we sample $n_m = n_\rho * n_{tasks}$ programs by mutating the population of training tasks, where $n_p$ is the desired number of policy samples per meta-iteration, and $n_{tasks}$ the total number of tasks in the population. We then evaluate each program on all tasks in the search set. For more details on this procedure, see Appendix A.2.

**Random sampling of programs with task relabelling**   For the random baseline, for each program line, we sample a primitive function at random from our DSL. We then sample its arguments given its type. When a variable which is of type grid is created, we end the program with probability 0.8, otherwise we continue writing more program lines. For each meta-iteration, we sample $n_m = n_\rho * n_{tasks}$ programs. We then evaluate each program on all tasks in the search set.

**ARC baselines**   We compare with baselines from the literature that report scores on the public ARC evaluation set. A direct comparison is sometimes difficult, as not all baselines apply their method to the full ARC evaluation set: Kolev et al. (2020); Alford et al. (2021) focus only on a subset, and Mirchandani et al. (2023) report performance on an aggregated set of both the train and eval splits. Ainooson et al. (2023) and Ferré (2021) do run a search procedure for a custom DSL on the full set, but do not train a network on the ARC train split. The best performing variant of Ainooson et al. (2023) does not involve training at all, so they can report performance numbers on the train set too. Since Ainooson et al. (2023) report the highest performance on the full ARC evaluation set, we choose it as our main baseline.

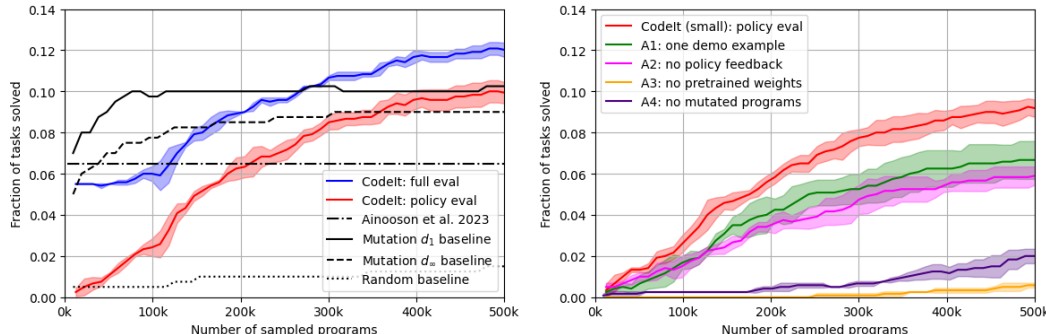

Figure 3: Left: Performance as function of number of sampled programs for CodeIt, mutation baselines, a random-search baseline, and baseline Ainooson et al. (2023). "CodeIt: full eval" considers all programs in the buffer as candidates, "CodeIt: policy eval" only considers programs sampled by the policy. Right: Ablations for a model with encoder context window size reduced from 1024 to 512. Solid lines indicate the mean, shaded area indicates standard deviation across three runs.

## 3.2 SETUP

We initialize our replay buffer with the 400 examples from the ARC training split and the associated solution programs provided by Hodel (2023). We also sample 3,038 programs as additional pre-training data via the mutation procedure outlined in Appendix A.2. We use the 400 ARC evaluation examples as our search set.

During the sampling stage of each meta-iteration, we use temperature sampling with temperature of 0.95, and sample up to $n_\rho = 24$ programs per task. This encourages exploration and, as a result, increases the diversity of data added to the replay buffer. Note that only a proportion of policy sampled and mutated programs are syntactically correct and, thus, are added to the buffer.

In each learning stage, we start by sampling a set of experiences from the buffer for training. This set always includes all tasks in the train split, because the ground truth programs are available for these tasks. It also includes tasks from the search set that have a perfect task demonstration performance. For some search set tasks, we may have multiple programs that reach maximum demonstration performance. In this case, we randomly select a program with weight $\frac{5}{|\rho|}$, where $|\rho|$ is the length of the candidate program. This effectively means we sample the shortest program most of the time.

For all other programs, resulting either from policy sampling or mutation, we prioritize more recent experiences. Each unique program can have multiple associated input-output pairs. For each program, we include corresponding input-output pairs with a probability of $1 - 0.99^i$, where $i$ is the meta-iteration in which the experience was added. For a full list of hyperparameters, see Table 3 in the Appendix.

## 3.3 MAIN RESULTS ON ARC EVAL SET

We report performance of CodeIt after sampling 500,000 programs, and that of various baselines, in Table 1. We also visualize the performance of CodeIt, the mutation baseline, and Ainooson et al. (2023) across meta-iterations in Figure 3. Our approach substantially outperforms both variants of Ainooson et al. (2023) and Mirchandani et al. (2023), and performs on par with GPT-4 (Gendron et al., 2023). As no other baseline scales to the full ARC evaluation set, our method represents the state-of-the-art on the ARC evaluation set.

We emphasize that the best-performing approaches (including the ARC Challenge 2022 winning solution of Hodel (2023), which we do not consider here) are ultimately all based on some variant of brute-force search. Our approach outperforms these despite the simplicity of the search procedure, compared to Ainooson et al. (2023). We argue that our method can achieve this result by successfully shifting the focus from brute force search to smart data acquisition and inter-task generalization, and is therefore closer in spirit to the idea of "fluid intelligence" that the ARC is designed to benchmark (Chollet, 2019).

| Method | ARC Train Set | ARC Eval Set | ARC Eval 412 |
|---|---|---|---|
| Ferré (2021) | 29 / 400 | 6 / 400 | - |
| Ainooson et al. (2023) MLE | 70 / 400 | 17 / 400 | - |
| Ainooson et al. (2023) brute force | 104 / 400 | 26 / 400 | - |
| Mirchandani et al. (2023) text-davinci-003 | 56 / 400[*] | 27 / 400[*] | - |
| Gendron et al. (2023) GPT-4 | - | - | 49 / 412[*] |
| Mutation $d_1$ | - | 41 / 400 | 36 / 412[*] |
| Mutation $d_\infty$ | - | 36 / 400 | 35 / 412[*] |
| Random sample | - | 6 / 400 | 7 / 412[*] |
| CodeIt: policy eval | - | 40 / 400 | 43 / 412[*] |
| CodeIt: full eval | - | **48 / 400** | 49 / 412[*] |

Table 1: Main results on ARC eval set. Our method outperforms all previous baselines. Evaluation metric is pass@3 by default, [*] indicates pass@1. To enable comparison to related work, we include pass@1 performance on the ARC Eval set with 412 examples. More details on this set and evaluation in Appendix A.3.

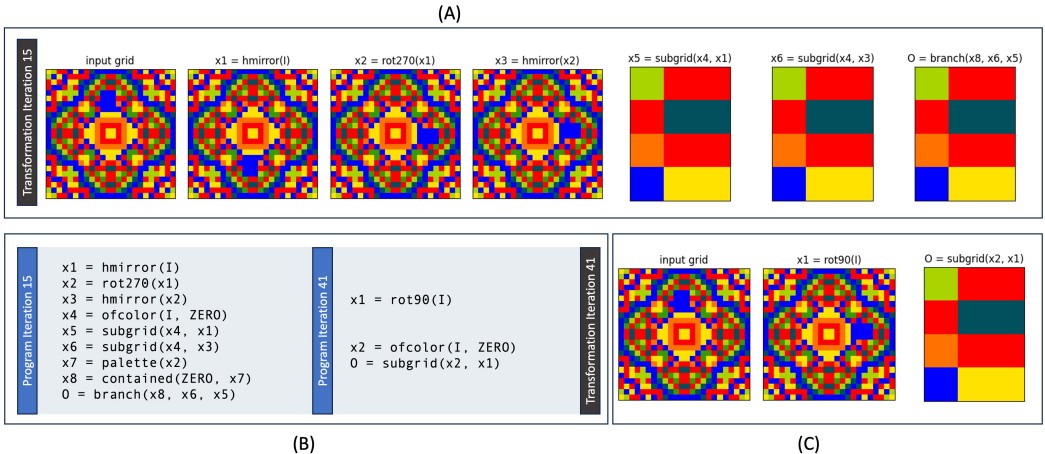

Figure 4: Program output traces for an example task. We show the found program at meta-iteration 15 (Panel A), the improved one found at meta-iteration 41 (Panel C) and program code for both (Panel B, right). We show the intermediate variable resulting from lines of code. At meta-iteration 41, CodeIt identified a much shorter solution program than the program at meta-iteration 15.

For the mutation baseline, we see a rapid performance increase followed by stagnation. CodeIt on the other hand has not stagnated, indicating that high-quality samples are found during search. Lastly, we address the balance between memorization and generalization. It is important to note that our approach does not solely rely on memorization. In the process, we mix in 3,038 random programs, at which point the mutation baseline has solved 5.5% of the tasks. This means that at most, 5.5% of solutions can be attributed to program memorization while abstracting between training and search set task representations. The majority of the performance stems from the model's ability to generalize, writing new and unseen programs.

To provide intuition, we show found programs for an example task in Figure 4. At meta-iteration 15 we obtain a longer program than necessary, but by meta-iteration 41, we see that the optimal program has been found, requiring only three lines to solve the problem. This also provides evidence that running the policy on tasks for which a solution program has previously been found is beneficial, as shorter programs are more likely to generalize from demonstration to test examples.

| Method | initial policy weights | # demo examples | # policy samples | Test perf. |
|---|---|---|---|---|
| CodeIt (small): policy eval | CodeT5 | $\leq 4$ | 24 | 37/400 |
| A1: One demo example | CodeT5 | 1 | 24 | 27/400 |
| A2: No policy feedback | CodeT5 | $\leq 4$ | 0 | 24/400 |
| A3: No pretrained weights | Random | $\leq 4$ | 24 | 2/400 |
| A4: No mutated programs | CodeT5 | $\leq 4$ | 24 | 8/400 |

Table 2: Ablation results, using a small version of CodeIt with context window size 512. The results highlight the importance of pretraining.

## 3.4 ABLATIONS

In Table 2, we ablate some of our design choices with the encoder context window reduced from 1024 to 512 due to computational constraints. Note that this means that for tasks with larger grid sizes fewer examples will be encoded. In all cases, we initialize the method with the ARC train set, and use the ARC evaluation set as the search set.

We first test the effect of the number of demonstration examples provided as input to the policy network (A1). Most tasks in the ARC dataset can only be solved when taking multiple demonstration examples into account, but there are many tasks that can be solved from a single example too. The primary objective of this ablation is to underscore the policy's capacity to derive intelligent insights from the input-output grids, rather than merely learning a sampling distribution over programs. By demonstrating enhanced performance with four demonstrations compared to one, we provide evidence supporting the policy's ability to learn a meaningful embedded representation of the task. It should be noted that the scale of this effect might be understated due to the reduced encoder context window. In our experimentation, running CodeIt with only one demonstration example resulted in diminished performance compared to when multiple examples are considered, reinforcing the importance of multiple demonstration examples in achieving optimal performance. Further, we pose that using a larger encoder context window would increase performance even further as less grid representations would be truncated.

In order to validate the efficacy of incorporating sampled programs during iterations, we execute an alternative scenario wherein the replay buffer remains static, and no additional policy samples are introduced (A2). In this variation, the policy continuously trains exclusively on the initial set of 400 programs and their 3,038 mutated counterparts. A noticeable decline in performance is observed in this setting. This outcome underscores the inference that the ongoing inclusion of samples from the evolving policy contributes to the enrichment of the training data quality.

We examine pretraining by reinitializing the policy's weights (A3). The modified policy takes an extended period to achieve non-zero performance and underperforms in comparison to its pre-trained counterpart. These results underscore the utility of beginning with a pre-trained model. The difference in performance is attributed to the pre-trained model's familiarity with Python and text comprehension. Since our programs are written in Python, starting with CodeT5, which already understands the language, expedites the learning process. The policy's adaptation to our custom DSL is also accelerated when beginning with pre-established foundational knowledge.

We also include a variation, *no mutated programs*, that skips the initial program mutation step, starting from just the 400 ARC training tasks. This model is slow to learn due to the reduced variety of data in the early stages of training, highlighting the importance of good initialization.

## 4 RELATED WORK

**Iterative policy improvement**   Iterative policy improvement procedures have been applied to many complex tasks, including game-playing (Silver et al., 2016; 2018), combinatorial problems (Laterre et al., 2019), question-answering (Zelikman et al., 2022), and program synthesis (Ellis et al., 2020; Fawzi et al., 2022; Gauthier & Urban, 2022; Butt et al., 2022). Expert iteration (ExIt) (Anthony et al., 2017; Silver et al., 2017; 2018) is one such class of approaches, consisting of a

policy-guided search stage that gathers new experiences, and a learning stage that improves the policy by imitation learning. Commonly used experts tend to be powerful and computationally intensive tree search algorithms such as Monte Carlo Tree Search (Kocsis & Szepesvári, 2006) and greedy search (Daumé et al., 2009). Reinforced self-training (ReST) (Gulcehre et al., 2023) shows that policy improvement is also possible without a strong search-based expert, but requires a pre-specified reward function for fine-grained filtering of generated data samples. In comparison to both ExIt and ReST, our approach is more straightforward: "search" consists of sampling from the policy, and sampled data are only filtered for syntactical correctness before learning.

A closely related work is the work of Haluptzok et al. (2022), which proposes a data generation pipeline for puzzle solving. Similar to our work, they propose an iterative approach: a language model policy generates solutions, then solutions are filtered on correctness using an interpreter. Key differences are in the way tasks are generated, and the way output programs are used. In their work, tasks are proposed by the language model, meaning most tasks are synthetic. In this work, we only use real task inputs, but use hindsight relabeling to further augment the set of input-output-program triplets in the buffer. Investigating whether these two ideas are complementary is an interesting area for future research.

A key challenge with iterative multi-stage procedures is to ensure all stages are working together. For example, if the policy overfits during learning, then the next search stage may yield less varied useful experiences, potentially leading to a negative feedback loop. Common techniques for mitigating this phenomenon include adding noise to policy output to promote exploration (Silver et al., 2016), injecting off-policy experiences into the replay buffer (Silver et al., 2016), or filtering experiences before training (Ye et al., 2020; Danihelka et al., 2021; Dong et al., 2023; Gulcehre et al., 2023). In this work, we introduce artificial experiences at the start of training via program mutation. After that, we only filter policy outputs for syntactical correctness, and require no explicit exploration incentive; instead we use hindsight relabeling of the target output to the true output to ensure that all correct policy outputs can be used in subsequent learning stages.

**Program synthesis**  The field of program synthesis is concerned with generating programs that solve specific problems (Gulwani, 2011), with many works aiming to train an agent capable of programming (Becker & Gottschlich, 2017; Kalyan et al., 2018; Bunel et al., 2018; Odena & Sutton, 2020). Notably, powerful language models have also been employed, demonstrating efficacy in automating code generation (Austin et al., 2021; Chen et al., 2021b; Fijalkow et al., 2021; Jain et al., 2022; Le et al., 2022). A subfield of program synthesis is programming-by-examples (Gulwani, 2011), where one aims to infer the program that produces a specific output given a given input, e.g. writing a function which correctly computes and returns the sum of its arguments. Recent works have successfully applied reinforcement learning approaches to this problem (Becker & Gottschlich, 2017; Ellis et al., 2019; Gauthier, 2022; Butt et al., 2022).

A closely related task is that of theorem proving (Han et al., 2021; Drori et al., 2022). Similar to programming-by-examples, the aim is to produce a series of formal statements that reaches a pre-specified goal, and to obtain formal verification of exactness of the resulting proof. In Polu & Sutskever (2020) and Polu et al. (2022), the authors reduce the problem to the one of finding, via a tree-search procedure, the right decomposition of a theorem into more easily provable sub-statements, whose proofs are obtained by querying an LLM trained with an expert iteration procedure. These works aim to showcase reasoning capability, similar to the goal of this work, but differ because the proposed methods rely on an advanced tree search.

**Abstraction and Reasoning Corpus**  Various works have applied program synthesis approaches to subsets of the ARC dataset. Xu et al. (2022) proposes to represent grids as graphs, and applies logical programs to the graph nodes, solving 63 of 160 tasks. Kolev et al. (2020) apply a Differentiable Neural Computer to ARC, solving 78% of tasks with grids of size $10 \times 10$ and smaller. Alford et al. (2022) applies DreamCoder (Ellis et al., 2020) and execution-guided program synthesis, solving 22 of 36 considered tasks. Park et al. (2023) first collects human feedback, then performs behavioral cloning for a subset of ARC tasks using a decision transformer (Chen et al., 2021a). However, none of these methods are applied on the full ARC evaluation set, typically due to poor scaling behavior.

The few works that do scale to the full evaluation set tend to solve each task in isolation. One of the most successful approaches on the private leaderboard of the yearly ARC challenge uses a breadth-

first search for a hand-designed DSL (Hodel, 2023). Ferré (2021) and Ainooson et al. (2023) both design a custom DSL and perform search for each task. Ainooson et al. (2023) obtains state of the art performance on the ARC evaluation set using brute-force search, solving 36 of 400 evaluation tasks. Mirchandani et al. (2023) demonstrate that a pretrained language model with custom tokenizer will output the correct grid after being shown multiple input-output pairs, obtaining correct solutions for 85 of 800 tasks. Wang et al. (2023a) further augment this approach by generating hypotheses in multiple rounds, although they only show performance on a subset of the ARC training set due to the high monetary cost of querying the language model. In this work, we design a scalable program synthesis approach that combines language models with the higher-level abstraction of a DSL. We also ensure that our approach benefits directly from generalization between tasks.

## 5 DISCUSSION

Leveraging a domain-specific language (DSL) allows CodeIt to reason at a higher level of abstraction, and to generate new tasks to learn from. However, this does mean a DSL must be available, or must be created first. A well-engineered DSL should support generalization and exploration through mutations and policy sampling, and designing one may not be straightforward. Additionally, one must have access to example programs written in this DSL for tasks of interest, in order to fine-tune pretrained language models. We address this issue via program mutation and hindsight relabeling, thus requiring only a small amount of expert programs to start training.

An important question is how much of our generalization capability is due to the model's ability to generate abstract representations, and how much is due to the inductive bias baked into the DSL. We observe that the mutation baseline obtains competitive performance, indicating the usefulness of a well-designed DSL. At the same time, our policy-based approach performs best overall, and we observe in practice that the policy samples new programs for the tasks of interest, indicating that it has learned to generalize between tasks. Both factors likely contribute to performance, and disentangling their effect is an interesting avenue for future research.

In iterative learning methods, there is a tradeoff between the quality and the quantity of experience generated by the sampling stage. Expert Iteration relies on a policy improvement operator: a powerful but computationally expensive sampling stage with a policy improvement guarantee. In this work, we opt for a cheaper sampling stage that gives no guarantee of (semantic) policy improvement, though does provide guarantee of syntactic correctness (due to filtering and hindsight relabeling). While CodeIt strikes an interesting balance—as our results show—it remains an open question what the optimal tradeoff might be.

## 6 CONCLUSION

In this work, we have presented an iterative learning method to solve problems in the Abstraction and Reasoning Corpus (ARC). Previous approaches either fail to scale to the full ARC evaluation dataset, or solve ARC tasks individually with brute-force search. We outperform these methods by achieving inter-task generalization using learned search. We show that a scalable search and fast learning enable our approach to quickly generate and learn from a diverse set of data, allowing experience to transfer. We demonstrate that our Code Iteration approach acquires the abstract reasoning capabilities required to solve ARC problems.

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

# A   Method and evaluation details

## A.1   CodeIt Algorithm

The pseudo code for the CodeIt procedure is portrayed in Algorithm 1.

**Initializing CodeIt**   Before we start the CodeIt procedure, we expand the training dataset using the first 3,308 mutated tasks from the mutation procedure (see Appendix A.2) used for the mutation $d_1$ baseline.

---

**Algorithm 1** CodeIt Algorithm

---

**Require:** Training set $D_{\text{train}}$, search set $D_{\text{test}}$, policy $Q$
**Ensure:** Finetuned policy $Q$, updated replay buffer $R$, $\rho^*$
1: $D_{\text{mutated\_train}} \leftarrow \text{EvolveTrainingTasks}(D_{\text{train}})$        ▷ Evolve training tasks
2: **Initialize** $R$ with $D_{\text{train}}$ and $D_{\text{mutated\_train}}$        ▷ Populate the initial replay buffer
3: **Initialize** $\rho^*$ as an empty set        ▷ Init set of programs that solve tasks in $D_{test}$
4: $D_{\text{sample}} \leftarrow \text{SampleFrom}(R)$        ▷ Sample tasks from the replay buffer
5: Train $Q$ on $D_{\text{sample}}$ for 20 epochs        ▷ Initial training of the policy
6: **for** meta_iter $= 1 \rightarrow 50$ **do**
7:      # Sampling stage
8:      **for** task in $D_{\text{test}}$ **do**
9:          $\{\rho\} \leftarrow Q(\rho|\{S_0, S_N\})$        ▷ Sample programs for test tasks
10:          **for** each $\rho$ in $\{\rho\}$ **do**        ▷ Test each inferred program
11:              **if** SyntacticallyValid$(\rho)$ **then**        ▷ If program outputs valid grids
12:                  Add $\{\rho, \{(S_0^{(i)}, \rho(S_N^{(i)})), \dots\}\}$ to $R$        ▷ Update the replay buffer
13:              **end if**
14:              **for** $(S_0^{(i)}, S_N^{(i)})$ in task **do**
15:                  **if** $\rho(S_0^{(i)}) = S_N^{(i)}$ **then**        ▷ Verify output
16:                      Add $\{(\rho, \text{task})\}$ to $\rho^*$        ▷ Update set of programs that solve tasks in $D_{test}$
17:                  **end if**
18:              **end for**
19:          **end for**
20:      **end for**
21:      # Learning stage
22:      $D_{\text{sample}} \leftarrow \text{SampleFrom}(R)$        ▷ Sample tasks from the replay buffer
23:      Train $Q$ on $D_{\text{sample}}$ for 1 epoch        ▷ Continual training of the policy
24: **end for**
25: **return** $Q, R, \rho^*$        ▷ Return the finetuned policy, updated buffer, and optimal programs set

---

## A.2   Program and Task Mutation

**Mutation procedure**   To grow a population of mutated programs with task demonstration inputs corresponding to the original training dataset, we follow the procedure outlined in Algorithm 3. This involves mutating a single task, which is described in Algorithm 2. The mutation is carried out with the hyperparameters $\phi_{\text{var}} = 0.25, \phi_{\text{arg}} = 0.5, \phi_{\text{func}} = 0.25$. With respect to naming notation, $d_1$ reflects a depth of 1, meaning we only mutate programs from the original training set, and $d_\infty$ reflects a depth of infinity, meaning we can mutate previously mutated programs.

The intuitive explanation of the mutation procedure for a single program is as follows. We pick a random line from a program (L2-3). We then replace either a function call with a function with similar output type (L4-7), or we replace an input argument in the function call (L8-11), or we replace the function call but leave its input variables the same (L12-14).

**Mutation baseline with task relabelling**   For our mutation baseline, we sample mutated programs using the mutation procedure outlined above. With task relabelling, for all the mutated programs in the evolved task population, we evaluate each program on the tasks in our search set.

---

**Algorithm 2** MutateProgram

---

**Require:** Replacement probabilities $\phi_{\text{var}}$, $\phi_{\text{arg}}$, $\phi_{\text{func}}$, program $\rho$
**Ensure:** $\rho'$
  1: **Initialize** $\rho' \leftarrow \rho$                                       ▷ Copy original program
  2: $l \leftarrow \text{RandomLineFrom}(\rho')$                                  ▷ Randomly select a line
  3: $p \sim U(0, 1)$
  4: **if** $p < \phi_{\text{var}}$ **then**                                       ▷ Variable mutation
  5:     $f' \leftarrow \text{SampleFunctionWithOutputType}(\text{GetTypeOfVariable}(l))$
  6:     $args' \leftarrow \text{SampleArgumentsForFunction}(f')$
  7:     Replace variable definition $f(args)$ in $l$ with $f'(args')$
  8: **else if** $p < (\phi_{\text{var}} + \phi_{\text{arg}})$ **then**            ▷ Argument mutation
  9:     $a \leftarrow \text{RandomArgumentFrom}(l)$
 10:     $a' \leftarrow \text{SampleTermOfType}(\text{GetTypeOfArgument}(a))$
 11:     Replace argument $a$ with $a'$
 12: **else**                                                                      ▷ Function mutation
 13:     $f' \leftarrow \text{SampleFunctionOfType}(\text{GetTypeOfFunction}(f))$
 14:     Replace function $f$ in $l$ with $f'$
 15: **end if**
          **return** $\rho'$

---

**Algorithm 3** EvolveTrainingTasks

---

**Require:** Initial population of training tasks $T_{\text{init}}$ (each task is a tuple $(\rho, \mathcal{E})$ where $\mathcal{E} = \{(S_0^{(i)}, S_N^{(i)}), \dots\}$, depth
**Ensure:** Updated task population $T'$ (initialized with $T_{\text{init}}$)
  1: $T \leftarrow T_{\text{init}}$
  2: $i \leftarrow 0$
  3: **while** $i < num\_samples$ **do**
  4:     **if** depth $= 1$ **then**
  5:         $(\rho, \mathcal{E}) \leftarrow \text{RandomSelectTask}(T_{\text{init}})$      ▷ Select from initial tasks
  6:     **else**
  7:         $(\rho, \mathcal{E}) \leftarrow \text{RandomSelectTask}(T)$              ▷ Select from current tasks
  8:     **end if**
  9:     $\rho' \leftarrow \text{MutateProgram}(\rho)$
 10:     $\mathcal{E}' \leftarrow \emptyset$                                       ▷ Initialize mutated task demonstration examples
 11:     **for** each $(S_0^{(k)}, \_) \in \mathcal{E}$ **do**
 12:         $S_N'^{(k)} \leftarrow \text{Execute}(\rho', S_0^{(k)})$
 13:         $\mathcal{E}' \leftarrow \mathcal{E}' \cup \{(S_0^{(k)}, S_N'^{(k)})\}$
 14:     **end for**
 15:     **if** $\text{AreValidGrids}(\text{GetAllOutputs}(\mathcal{E}'))$ **then**
 16:         $T' \leftarrow T' \cup \{(\rho', \mathcal{E}')\}$                      ▷ Add new task to the population
 17:     **end if**
 18:     $i \leftarrow i + 1$
 19: **end while**
          **return** $T'$

---

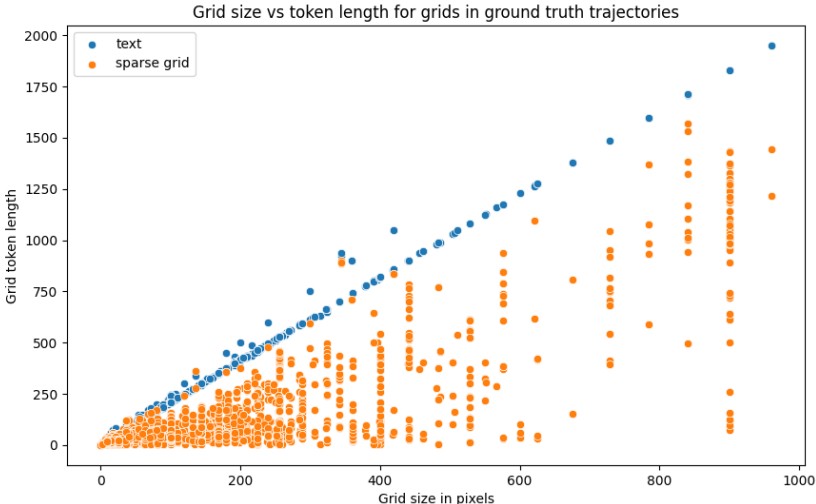

Figure 5: Grid size versus token count for the ARC training data.

## A.3 TASK REPRESENTATION

**Grid representation**   We use a compressed grid representation, mainly to reduce the number of tokens needed to represent each grid. We do not use a custom tokenizer. A visualization of the number of tokens is shown in Fig. 5, showing that in almost all cases, the sparse grid representation we use leads to a reduction in the number of needed tokens, especially for larger grid sizes.

**Truncation**   We truncate our task demonstration tokens and program tokens such that these sequences fit in our predefined encoder and decoder context windows. For the task demonstration examples, we first order by grid size and divide the encoder context window into two equally sized sections. For task demonstration inputs, we first encode input grids to text as above and then we tokenize using the standard text tokenizer. We truncate these tokens at half the size of the encoder context window. We do the same for the task demonstration outputs and with the exception of also adding an end of sequence token. As a result, even though we aim to show the policy up to four task demonstration examples, large grids will be cut-off. For programs, we tokenize directly using the standard text tokenizer and truncate at the decoder context window size.

## A.4 ARC EVALUATION

Different works use different evaluation procedures to report performance on the ARC evaluation set. We describe two common evaluation settings in more detail below. Unless mentioned otherwise, we always use the first procedure, "ARC Eval Set".

**ARC Eval Set**   This setup is intended as close as possible to the evaluation procedure described by Chollet (2019). Baselines Ferré (2021), Ainooson et al. (2023) follow this procedure, and it is our default setting as well.

The ARC eval set consists of 400 tasks, some of which contain multiple test examples. Common procedure is to report pass@3 performance, meaning the top 3 solutions are selected according to demonstration task performance. If there are ties, we favor the shorter program, under the assumption that shorter programs are more likely to generalize. We then run these programs on all test examples for the task. In some cases, there are multiple test examples per task. We call the task "solved" if all output grids are correct.

**ARC Eval 412** This setup is designed to match Gendron et al. (2023). Instead of calling a task with multiple test examples solved if all test outputs are correct, distinct tasks are created - one per test example. This results in a set of 412 evaluation tasks with one test example each. Furthermore, Gendron et al. (2023) uses pass@1, rather than pass@3: only one solution per task is evaluated, and the task is considered solved if the output is correct.

## B  EXPERIMENT DETAILS

### B.1  RESOURCES

Main experiments were run for a maximum of 80 hours on a NVIDIA A100 80GB. Ablation experiments were run for a maximum for 80 hours on a NVIDIA Tesla V100 32GB.

### B.2  HYPERPARAMETER TUNING

**Dataset** The ARC benchmark does not contain a validation split. Hence, we use part of the ARC train split for validation during the hyperparameter tuning. In particular, this validation set is the search set that the sampling stage uses as described in 2.2. With this setup we avoid overfitting the hyperparameters to the ARC evaluation split. We choose the split such that $\mathcal{D}_{\text{train}}$ and $\mathcal{D}_{\text{valid}}$ contain roughly equally difficult programs by sampling based on program length: $\mathcal{D}_{\text{train}}$ contains 80% of 2-line programs, 80% of 3-line programs, and so on. This results in 311 examples in $\mathcal{D}_{\text{train}}$ and 89 examples in $\mathcal{D}_{\text{valid}}$.

**Experiments on validation set** In these experiments, we initialise our replay buffer with the 311 $\mathcal{D}_{\text{train}}$ examples, and our search set consists of the 89 $\mathcal{D}_{\text{valid}}$ examples. The aim of these experiments is to find optimal hyper-parameters for search and training. A list of our tuned hyperparameter values and their description is shown in Tab. 3

### B.3  HYPERPARAMATERS CHOSEN ON INTERNAL VALIDATION SET

We optimized these parameters on our custom validation set before applying CodeIt to ARC eval.

| CodeIt stage | Param | Value | Description |
|---|---|---|---|
| Sampling | $n_\rho$ | 24 | no. policy samples $\rho$ per task per meta-iteration[1] |
| | $n_m$ | 3,038 | no. mutated task samples for replay buffer initialisation[1] |
| | $\tau$ | 0.95 | sampling temperature |
| Learning | $n_\epsilon$ | 1 | no. train epochs per meta-iteration $i > 0$ |
| | $n_{\epsilon_1}$ | 20 | no. (pre-)train epochs in first meta-iteration $i = 0$ |
| | $lr$ | $5e-5$ | learning rate |

Table 3: Table of hyperparameters.

### B.4  DOMAIN SPECIFIC LANGUAGE

We adopt the domain specific language (DSL) of Michael Hodel, made available on GitHub: https://github.com/michaelhodel/arc-dsl. This DSL was designed based on the training set: the (human) designer did not peek at the evaluation set. This is what allows us to run search on ARC eval here. Using a DSL designed for the eval tasks would be cheating, as we would benefit immensely from human insights captured in the primitives. On the other hand, it may mean that some ARC eval programs are not solvable with the current DSL.

---

[1]Note that no. samples here refers to policy and mutation samples before filtering for syntactic correctness.

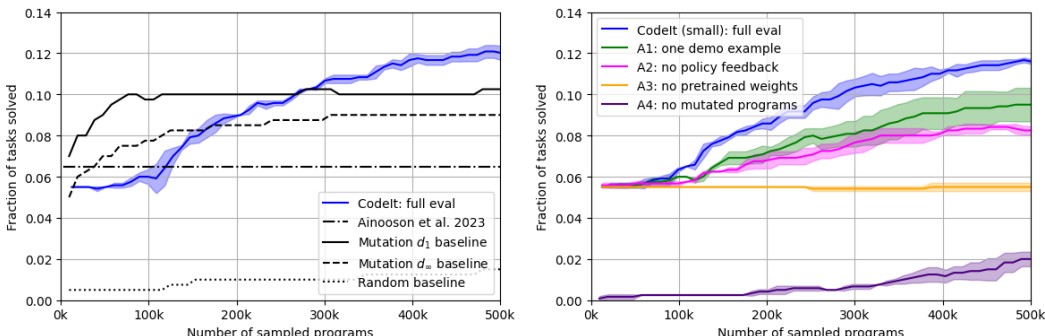

Figure 6: Pass@3 performance versus number of samples, where we consider a task solved if either mutation or the CodeIt policy identified a solution. This means CodeIt starts from $5.5\%$ tasks solved. Left: main results and comparison to baselines. Right: ablations.

The DSL is implemented in https://github.com/michaelhodel/arc-dsl/blob/main/dsl.py. It contains many basic grid manipulation operations, such as rotations (`rot90`, `rot180`, `rot270`), mirroring (`dmirror`, `hmirror`, `vmirror`), resizing (`downscale`, `upscale`), or concatenation (`hconcat`, `vconcat`). It also contains functions that perform counting, for example `numcolors` counts the number of colors occurring in an object or grid. For some ARC tasks, identifying the foreground objects and determining how these objects interact is an effective strategy for human test-takers. Therefore, some functions also apply to "objects", which are patches of the same color that stand out from the background. To extract these, the function `objects` returns the set of foreground objects, i.e. those that have a different color than the most common color, assumed to be the background. For a complete list of primitives and their description, we refer the reader to the aforementioned Github page.

Michael Hodel was kind enough to also provide hand-designed solution programs for all training tasks in https://github.com/michaelhodel/arc-dsl/blob/main/solvers.py. Some programs are highly complex: for some of the more challenging ARC tasks, we see solutions consisting of up to 58 lines of code (`solve_b775ac94`). We use these 400 solution programs to kickstart CodeIt training.

## C SUPPLEMENTARY RESULTS

In the main text Figure 3, we showed performance of CodeIt based on the policy performance. That is, we only consider programs sampled from the policy for evaluation, and effectively ignore that the mutation baseline has already identified candidate solutions – as these solutions are in the replay buffer, the CodeIt policy will learn about these as we train.

However, we could also combine the best of both worlds, by including samples from the initial mutation step during evaluation. That is, if a mutated program from the buffer obtains better demonstration task performance than any of the policy samples, we take this as the solution program during evaluation. In Figure 6 on the left, we show the resulting pass@3 performance as a function of number of samples. The immediate effect is that CodeIt starts from $5.5\%$ of tasks solved, as the buffer already contains solutions for a number of tasks from the initial mutation step. Secondly, we observe that this allows us to outperform the mutation baseline, as CodeIt finds solutions to tasks that the mutation baseline does not solve.

In Figure 6 on the right, we show the pass@3 performance for all ablations. All runs start from a set of mutated programs, except A4 "no mutated programs", which then also starts from $0\%$ tasks solved. A striking observation is that A3, which does not use pretrained weights, completely stagnates, indicating that despite the domain shift there is a clear advantage to using a model pretrained on code synthesis. For A1 and A2, behavior is similar to Figure 3 in the main text: showing only one

| Method | Number of tasks solved |
|---|---:|
| CodeIt only | 14 |
| Mutation $d_1$ only | 8 |
| Mutation $d_\infty$ only | 4 |
| CodeIt ∩ Mutation $d_1$ | 23 |
| CodeIt ∩ Mutation $d_\infty$ | 22 |
| Mutation $d_1$ ∩ Mutation $d_\infty$ | 27 |
| CodeIt ∩ Mutation $d_1$ ∩ Mutation $d_\infty$ | 17 |

Table 4: ARC evaluation tasks solved per method, plus unique number of tasks solved. The top group of three rows show how many programs were identified by a method, but not by the other two. The center group of three rows show which tasks were solved by two of the three methods. The final row shows tasks solved by all three methods.

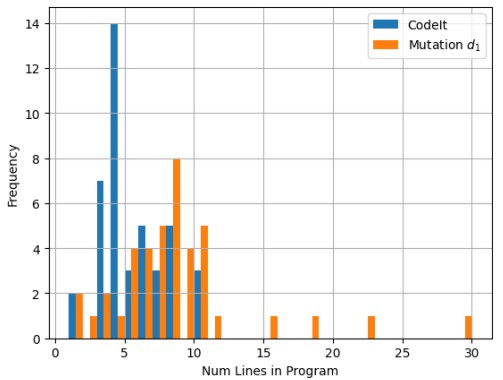

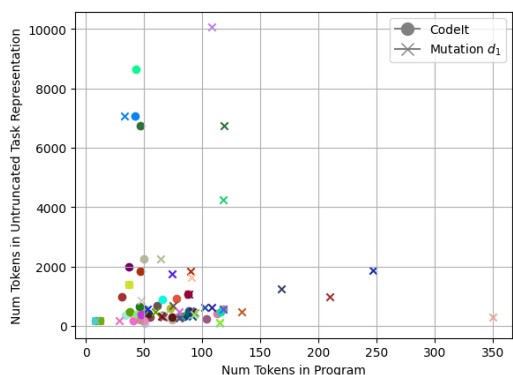

(a) Histogram of number of lines for tasks where both CodeIt and Mutation produced solutions. CodeIt (in blue) typically produces shorter programs than the Mutation baseline (in orange).

(b) Number of task representation tokens vs number of program tokens. Colors represents the different tasks. We see no obvious correlation between task representation length and program length.

Figure 7: Shortest programs for solved ARC evaluation tasks for CodeIt and the Mutation baseline.

demonstration example (A1) harms performance, and (A2) using only the initial set of 400 ground truth + 3,038 mutated samples results in worse performance still.

# D   PROGRAM ANALYSIS

## D.1   PROGRAM LENGTH

We compare the programs found using our mutation $d_1$ baseline, mutation $d_\infty$ baseline and the best performing of the three CodeIt runs. Table 4 displays the number of ARC evaluation tasks uniquely solved by each method and the tasks which are solved by multiple methods. Overall, CodeIt solves 42/400 tasks, 14 of which neither of the mutation baselines solve. In Figure 7, we select the shortest program that solves an evaluation task for CodeIt and our mutation $d_1$ baseline, computing the program length and task representation size. Note that CodeIt has an encoder context window size of 1024 and so any tasks which having representations of more than 1024 tokens have been truncated. Overall, CodeIt finds shorter programs as shown in 7a. Further, for the same task, CodeIt more often finds shorter programs than our mutation $d_1$ baseline, as shown in 7b where each color represents a different task. Interestingly, CodeIt does solve some tasks with very large task representations, suggesting in some cases a truncated task representation provides sufficient information to solve the task.

## D.2 COMPARISON WITH MUTATION SOLUTIONS

In Tables 5 and 6, we show a subset of solution programs for ARC eval tasks solved by both CodeIt and our mutation $d_1$ baseline. We select tasks where the shortest programs differ between the two methods. CodeIt programs appear more concise and use different primitives. Out of the 23 tasks that are solved by both methods, there are 18 shortest programs where method output is different. CodeIt produces a shorter program in 16 out of these 18 cases. The 2 longer CodeIt programs are displayed in Table 6.

The Mutation baseline often includes redundant lines, for example producing a line `x4 = vconcat(x2, x3)` that is not used to produce the final solution. Of course, these can be filtered out in hindsight: whenever a variable is unused, we do not execute the line. However, for many programs, CodeIt produces a program that is qualitatively better: the solution is less complex, and contains fewer lines overall.

## D.3 CODEIT SOLUTIONS AND OPTIMIZATION OVER META-ITERATIONS

This section shows the evolution of solutions generated by CodeIt across meta-iterations. Although CodeIt is not explicitly optimizing for shorter solutions, a trend towards more concise solutions emerges in later meta-iterations. Specifically, in 81% (standard error 2.55) of solved tasks, the first solution program proposed by CodeIt is refined to a more compact form in a later meta-iteration. Furthermore, for 63.1% (standard error 1.8) of programs that have at least three solutions, the shortest solutions are predominantly found in the final 50% of iterations since the solution is first found. This suggests a consistent improvement in solution efficiency over time. However, we observe that some solutions also increase in length. Table 7 shows a selection of tasks where compression occurs. The compression usually consists of removing redundant lines from the program and/or using a more directly applicable primitive.

| CodeIt | Mutation $d_1$ |
|---|---|
| ```
x1 = ofcolor(I, ONE)
x2 = lbind(shift, x1)
x3 = mapply(x2, x1)
O = underfill(I, TWO, x3)
``` | ```
x1 = rot180(I)
x2 = ofcolor(I, ONE)
x3 = ofcolor(x1, ONE)
x4 = neighbors(ORIGIN)
x5 = mapply(neighbors, x4)
x6 = lbind(shift, x3)
x7 = apply(x6, x5)
x8 = lbind(intersection, x2)
x9 = argmax(x7, x8)
O = underfill(I, TWO, x9)
``` |
| ```
x1 = replace(I, EIGHT, ZERO)
x2 = compress(x1)
O = trim(x2)
``` | ```
x1 = asindices(I)
x2 = fgpartition(I)
x3 = rbind(greater, TWO)
x4 = compose(x3, size)
x5 = sfilter(x2, x4)
x6 = totuple(x5)
x7 = apply(color, x6)
x8 = apply(center, x6)
x9 = pair(x7, x8)
x10 = fill(I, ZERO, x1)
x11 = paint(x10, x9)
x12 = rbind(greater, ONE)
x13 = compose(dedupe, totuple)
x14 = chain(x12, size, x13)
x15 = sfilter(x11, x14)
x16 = rot90(x15)
x17 = sfilter(x16, x14)
O = rot270(x17)
``` |
| ```
x1 = bottomhalf(I)
x2 = cellwise(x1, I, THREE)
O = switch(x2, ONE, TWO)
``` | ```
x1 = tophalf(I)
x2 = bottomhalf(I)
x3 = ofcolor(x1, ZERO)
x4 = ofcolor(x2, ZERO)
x5 = intersection(x3, x4)
x6 = astuple(FOUR, FIVE)
x7 = canvas(THREE, x6)
O = fill(x7, ZERO, x5)
``` |
| ```
x1 = hmirror(I)
x2 = ofcolor(I, THREE)
x3 = subgrid(x2, x1)
O = replace(x3, THREE, EIGHT)
``` | ```
x1 = hmirror(I)
x2 = vmirror(I)
x3 = ofcolor(I, THREE)
x4 = subgrid(x3, x1)
x5 = subgrid(x3, x2)
x6 = palette(x4)
x7 = contained(ONE, x6)
O = branch(x7, x5, x4)
``` |

Table 5: Selection of shortest programs for ARC evaluation tasks solved by CodeIt (left) and the Mutation $d_1$ baseline (right) for which CodeIt program is shorter.

| CodeIt | Mutation $d_1$ |
|---|---|
| ```
x1 = rot90(I)
x2 = ofcolor(x1, ONE)
O = crop(x1, ORIGIN, TWO_BY_TWO)
``` | ```
x1 = rot90(I)
O = crop(x1, ORIGIN, TWO_BY_TWO)
``` |
| ```
x1 = hmirror(I)
x2 = rot180(I)
x3 = ofcolor(I, ZERO)
O = subgrid(x3, x1)
``` | ```
x1 = rot90(I)
x2 = ofcolor(I, ZERO)
O = subgrid(x2, x1)
``` |

Table 6: All shortest programs for ARC evaluation tasks solved by CodeIt (left) and the Mutation $d_1$ baseline (right) for which the CodeIt program is longer.

| Long program | Short program |
|---|---|
| **Task key: 1a2e2828** | |
| *Longest (Iteration 15)* | *Shortest (Iteration 45)* |
| ```
x1 = objects(I, T, T, T)
x2 = first(x1)
x3 = subgrid(x2, I)
x4 = leastcolor(I)
x5 = leastcolor(x3)
x6 = ofcolor(I, x4)
x7 = ofcolor(x3, x5)
x8 = intersection(x6, x7)
x9 = canvas(x4, UNITY)
O = fill(x9, x5, x8)
``` | ```
x1 = objects(I, T, F, F)
x2 = argmax(x1, height)
x3 = color(x2)
O = canvas(x3, UNITY)
``` |
| **Task key: f0df5ff0** | |
| *Longest (Iteration 24)* | *Shortest (Iteration 47)* |
| ```
x1 = objects(I, T, F, T)
x2 = colorfilter(x1, ONE)
x3 = sfilter(x2, square)
x4 = compose(backdrop, inbox)
x5 = mapply(x4, x3)
O = underfill(I, ONE, x5)
``` | ```
x1 = ofcolor(I, ONE)
x2 = mapply(neighbors, x1)
O = underfill(I, ONE, x2)
``` |
| **Task key: e133d23d** | |
| *Longest (Iteration 27)* | *Shortest (Iteration 50)* |
| ```
x1 = lefthalf(I)
x2 = righthalf(I)
x3 = ofcolor(x1, ZERO)
x4 = ofcolor(x2, ZERO)
x5 = intersection(x3, x4)
x6 = astuple(THREE, THREE)
x7 = canvas(TWO, x6)
x8 = fill(x7, ZERO, x5)
O = replace(x8, SIX, TWO)
``` | ```
x1 = vmirror(I)
x2 = lefthalf(I)
x3 = righthalf(I)
x4 = cellwise(x2, x3, TWO)
O = replace(x4, SIX, SIX)
``` |
| **Task key: e0fb7511** | |
| *Longest (Iteration 30)* | *Shortest (Iteration 50)* |
| ```
x1 = objects(I, F, F, T)
x2 = colorfilter(x1, ZERO)
x3 = sfilter(x2, square)
x4 = sizefilter(x3, ONE)
x5 = merge(x4)
x6 = fill(I, THREE, x5)
x7 = merge(x3)
x8 = fill(x6, EIGHT, x7)
O = switch(x8, EIGHT, ZERO)
``` | ```
x1 = objects(I, T, F, T)
x2 = mfilter(x1, square)
x3 = fill(I, EIGHT, x2)
O = switch(x3, EIGHT, ZERO)
``` |
| **Task key: 195ba7dc** | |
| *Longest (Iteration 36)* | *Shortest (Iteration 49)* |
| ```
x1 = lefthalf(I)
x2 = righthalf(I)
x3 = ofcolor(x1, SEVEN)
x4 = ofcolor(x2, SEVEN)
x5 = combine(x3, x4)
x6 = intersection(x3, x5)
x7 = fill(x1, ONE, x5)
x8 = replace(x7, SEVEN, ONE)
O = replace(x8, EIGHT, TWO)
``` | ```
x1 = lefthalf(I)
x2 = righthalf(I)
x3 = cellwise(x1, x2, ONE)
O = replace(x3, SEVEN, ONE)
``` |

Table 7: Selection of longest (left) and shortest programs (right) for ARC evaluation tasks solved by CodeIt.

