# OpenReview forum: "CodeIt: Abstract Reasoning with Iterative Policy-Guided Program Synthesis"
_ICLR.cc/2024/Conference — Submitted to ICLR 2024_

### Official Review · Reviewer_NHnw · 2023-10-23

**Soundness:** 4 excellent
**Presentation:** 4 excellent
**Contribution:** 4 excellent
**Rating:** 6
**Confidence:** 5

**Summary:**

This paper describes a new program synthesis approach for solving the Abstraction and Reasoning Corpus (ARC). The authors take a 220m parameter T5 code pretrained language model, fine-tune it on handwritten solutions to the 400 training set tasks using a DSL designed for ARC as well as randomly mutated solutions to these tasks, and then train a policy by iteratively attempting to solve tasks and adding hindsight-relabelled solutions to a buffer of tasks to train the policy on. Their approach solves 40/400 tasks on the evaluation split, and improves over time.

**Strengths:**

The approach is excellent: well-designed, simple in principle yet careful in the details, and seems to be tuned well. The design decisions are carefully explained and justified, and the whole procedure is written out well in a way that is easy to follow.

This is a natural approach for ARC that has not been tried yet, and it is really exciting to see the results the authors have generated. Many ARC papers, as the authors note, are not good enough to use as a baseline, so the fact that this work succeeds in attempting to solve tasks from the evaluation set is a noteworthy accomplishment. Moreover, the approach is not simply brute-force search, and has the potential for much better performance if various parts of the system are tweaked and improved.

The related work is good. The ablation experiments are good. The discussion section is good. Really, it is a very nicely written paper.

**Weaknesses:**

- Overall, the writing could use general revisions for clarity, mainly on little details of wordings rather than high level changes.
- There is no mention of code being available or made open source.
- The authors only evaluate their approach on ARC, even though it could be in principle compared to synthesis approaches in other domains. Due to the difficulty and uniqueness of ARC, I think only evaluating on ARC is more than sufficient, but showing performance on another domain would help elucidate what about the algorithm is working and not working as expected.
- There is limited insight into understanding the capabilities of the model. For example, the paper does not convey well what exactly the model is improving at over time, or how competent the model is at generating syntactically correct code or using the full range of DSL operators once the fine-tuning stage is completed.

Overall, despite the well-written nature of the paper and good design of the approach. I find it hard to have a good grasp of how well the approach actually worked. 40/400 tasks seems a bit low given the technique, and there's not much space in the paper devoted to understanding what exactly is going on with the model. The ablations help a bit. I think the best way to get a sense of this would be to have a link to a page which shows the tasks and generated solutions the model discovers over the course of training, and when each one was found. This would really help understand how well the approach is working. In addition, seeing some of the tasks it fails to solve, which we might expect it to solve (e.g. if the DSL can solve it in a few operators), would help too. While this might make the approach look less impressive, it would really increase the quality of the paper.

**Questions:**

Questions
- Why don't you evaluate on the hidden test set? Even if you do not beat the state of the art, it would be good to know the performance.
- I don't understand how hindsight relabelling works with the mutation baseline in section 3.1 / appendix A.2 — for searching via mutation, it seems like you just need to store the program and the inputs into the buffer, no need to store the outputs?
- Your main approach that combines mutation and policy just uses mutation for generating the data during fine-tuning, is that right? There's none of the iterative random mutation search to try to solve tasks?
- Do you have any evidence that the pretrained model is comfortable using the DSL after pretraining? Or does it still struggle to use the correct input args, etc? not sure how best to convey this, but any information on this could help understand the model's capabilities better.
- I'm wondering if a grid array number encoding works better for training, because it's more like how T5 has seen things encoded before, instead of a brand new encoding it doesn't really understand. did you try this at all, or just stick to the computationally cheaper representation?
- You mention that including the mutated programs is a form of regularization, but I would prefer to call it data augmentation. Both imply preventing overfitting, but based on my understanding of the term, adding more data doesn't really regularize a model per se.
- What % of the time do the sampled programs execute successfully?
- 40/400 eval is not the state of the art on eval, is it? I think brute force approaches have solved a high fraction of the eval set — can you clarify? For example, I recall the original ARC kaggle competition winner saying they solved at least a hundred on the evaluation set using their DSL: https://github.com/top-quarks/ARC-solution/tree/master

Suggestions (feel free to ignore if you feel otherwise, and no need to discuss in rebuttal)
- I think the ARC example in figure 1 should be more complicated. This would both help audiences unfamiliar with ARC better understand the nature of the dataset, and would make the object encoding scheme easier to understand when looking at how the input is encoded — right now, the object encoding for Figure 1 has a lot of 0's, 1's, and 2's, which makes it hard to quickly understand how the example maps to the input.
- My suggested improved title: Solving the Abstraction and Reasoning Corpus with Iterative policy-guided program synthesis. Justification: you're only evaluating on ARC, and calling your approach "abstract reasoning" is only true in the sense that it's applied to the abstraction and reasoning corpus. I don't think your LLM is doing much abstraction or reasoning, in the sense of forming new abstractions itself, or chaining multiple steps of thinking together to arrive at its solution (debatable though depending on your NN philosophy)
- It might be good to include more examples of the DSL operators, so readers can have a rough sense of what your LM is generating without having to fully read Hodel's DSL explanation. Figure 4 is useful for this, but maybe you can show an example earlier on for this.
- Some details to clarify in the main text:
    - the mutated programs are evaluated on the inputs of the program it mutated off of to generate outputs
    - some more details on fine-tuning: how many mutated programs, how many epochs of training.
    - you mention weighing training on the handwritten solutions and solved tasks more often, so that you don't forget them, but is this explained further, or in the pseudocode in the appendix? if not, they should be included (maybe I missed it)
    - I would rewrite the last sentence of the "sampling stage" paragraph in section 2.2 to be clearer that you sample n_p times total.
- I would be careful not to use the word "test", as it might be mistaken for ARC's hidden test set. for example, in Table 2, I would call it Eval performance, not Test performance.
- You have a "policy only" baseline in Figure 3, which should be described or at least listed in the "baselines" section. In particular, I am confused whether "policy only" means that the initial fine-tuning doesn't have the mutated tasks, or if the "policy+mutation" means that you're searching for new tasks solutions at each iteration with both with the policy and via random mutations.
- Clarify that the solution programs were also written by Hodel — they deserve credit for that!

---

> ### Author Response · Authors · 2023-11-20
>
> We thank the reviewer for their in-depth comments and constructive questions. We highlight updates to the paper based on this review in ```purple```.
>
> > There is limited insight into understanding the capabilities of the model.
>
> We included additional analysis and results to provide more intuition for why the method works. For example, we include a comparison between solutions identified by the mutation baseline and CodeIt in Appendix D.1 and D.2, and analyze how CodeIt solutions change over time in Appendix D.3. We observe, for example, that in 81% of tasks for which CodeIt finds a solution, a shorter solution is identified in a later meta-iteration. This indicates that the method may use insights gained in some tasks to improve its solution for other tasks.
>
> > Why don't you evaluate on the hidden test set?
>
> Although this evaluation can provide another datapoint, this set is not available to the public, making it difficult to reproduce results. Not many published works report this performance, and it can only be obtained by submitting code to Kaggle or the ARC challenge. We did consider it however, and may do so at a later stage.
>
> > I don't understand how hindsight relabelling works with the mutation baseline in section 3.1 / appendix A.2
>
> For the mutation baseline, we perform task relabeling, an exhaustive form of hindsight relabeling similar to [Gauthier 2022]. We run each program produced by mutation on *all* tasks, not just the task corresponding to the expert program we're mutating. Mutating is semi-random, so it sometimes results in a lucky hit, where we unexpectedly solve a different task. We do not perform this task relabeling procedure for CodeIt.
>
> > Do you have any evidence that the pretrained model is comfortable using the DSL after pretraining?
>
> We do not test this explicitly, but we can infer this from the ablations in Figure 3, where we see that CodeIt without pretraining is unable to improve performance.
>
> > I'm wondering if a grid array number encoding works better
>
> We picked our encoding scheme to minimize encoding length, as this would enable fitting this in the (limited) LLM context window. There is also some evidence that object-based representations of ARC tasks are easier to work with for LLMs [Xu 2023]. But it is possible that the best encoding for ARC tasks indeed depends on the base model and its pretraining data.
>
> > You mention that including the mutated programs is a form of regularization, but I would prefer to call it data augmentation.
>
> We agree, this is a better choice of words. We updated the text.
>
> > What % of the time do the sampled programs execute successfully?
>
> We did not track this number, as we filtered both programs that did not execute correctly and duplicate programs, so we have no way to distinguish between the two. After training we get about 5,000 programs for 24*400 = 9,600 sampled tasks, so it is likely that more than 50% of programs execute successfully.
>
> > I recall the original ARC kaggle competition winner saying they solved at least a hundred on the evaluation set using their DSL
>
> This is correct, this method should give 129 correct programs. However, the DSL was designed by looking at the eval set. This means a human designer explicitly built priors into the DSL that would help with solving ARC evaluation programs. Michael Hodel first and foremost looked at the training set when designing the DSL, and thus stayed more true to the spirit of the ARC challenge as described by [Chollet 2019].
>
> > It might be good to include more examples of the DSL operators .. Clarify that the solution programs were also written by Hodel — they deserve credit for that!
>
> Fully agreed, we did acknowledge this in Section 3. We added more details about the DSL and links to the GitHub repo of Hodel in Appendix B.4 - and explicitly mention that he built the 400 solution programs we used to kickstart CodeIt.
>
> Other comments:
>  - we added more details about the hyperparameters and how we tuned these.
>  - we added a description for the CodeIt variations in Figure 3, also see the main response above for details on the changes.
>
> References:
> - [Gauthier 2022]: "Learning Program Synthesis for Integer Sequences from Scratch", Gauthier & Urban, 2022
> - [Xu 2023]: "LLMs and the Abstraction and Reasoning Corpus: Successes, Failures, and the Importance of Object-based Representations", Xu et al., 2023

---

> > ### Comment · Reviewer_NHnw · 2023-11-22
> >
> > Thank you for your comment and for addressing my questions!

---

### Official Review · Reviewer_xY18 · 2023-10-31

**Soundness:** 3 good
**Presentation:** 3 good
**Contribution:** 3 good
**Rating:** 6
**Confidence:** 4

**Summary:**

The paper proposes CodeIt, a program synthesis method that leverages learned prior for sampling and learning for ARC problem solving. To solve an ARC instance, the method uses a code-based pretrained network to sample program variants and used the augmented data to retrain the network for final program prediction. With good implementation, the method achieves good performance compared to previous state of the art, and the ablation studies show the contribution of proposed components.

**Strengths:**

The method is essentially a simplified version of DreamCoder, the dream part in particular. The pretrained code-based network serves as the learned prior, and the sampling stage basically tries to augment the input-output-program triplets such that they could be used for fortifying the network, providing locally diversified data for additional training. This is a pretty intuitive way of attempting the problem. However, some questions still persist, see below.

**Weaknesses:**

As the method still largely relies on the data it samples, I wouldn't be surprised to see it becomes better than previous methods. However, I'm also interested in hearing where the limit of the simple mutation baseline is: if you more extensively sample mutations and in the extreme case, the mutations cover all your newly sampled data, would your method becomes inferior? From my perspective, your method might only be better than the mutation baseline, because your pretrained policy network serves as some smart prior, and to get exactly the programs your policy samples, random mutation might simply take more than (less efficient). However, using a unverified prior would also risk data coverage, meaning that it might not cover as much data as the random mutation method. In this sense, the random mutation, while inefficient, when taken to the extreme, could possibly be better.

As the method is basically data-driven, but smartly, I would be expecting to see ablation on the amount of sampled data, rather than the context length. How would the model perform if you reduce the number of samples, or even better, can you show the curve of perf vs. num of samples per task? If you decrease that number, your performance might not be better than the mutation baseline.

Another problem regarding ARC in general is evaluation. There has been no equal footing as for the number of data one could use. I note that in your method, you have incorporated quite a lot of sampled programs, and I seriously doubt what would happen when the newly sampled programs are used by other methods. Also a pretrained CodeT5+ is used, which already sees quite a lot of data. In the extreme, one would like to sample the space as much as possible and feed them to a model, saving all the trouble in modeling.

Experiments only on ARC are not necessarily sufficient to show the superior of the method. I knew of the Raven matrices that also stress abstract reasoning, and the RAVEN dataset has similarly structured data and a much simplified program structure. Would it be possible to show similarly improved performance on this task?

One thing I've been thinking about ARC is that the community has been doing program search on a fixed DSL for a while. Would it be possible to jointly search over the DSL space, maybe starting simple such as using a mutation method for the DSL space?

**Questions:**

See weaknesses. A lot of the questions may not be properly answered under the current climate, but please try to.

---

> ### Author Response · Authors · 2023-11-20
> **Response to reviewer xY18**
>
> We thank the reviewer for their positive assessment of our work and their insights. Changes to the paper related to his review are listed in ```orange```.
>
> > I’m also interested in hearing where the limit of the simple
> mutation baseline is: if you more extensively sample mutations and in
> the extreme case, the mutations cover all your newly sampled data, would
> your method becomes inferior?
>
> Our experiments were indeed designed to maintain data-parity. If one were
> to increase the number of sampled mutations, one should also increase the
> number of policy samples to ensure fair comparison. It would be unfair
> to allow the mutation to sample new programs indefinitely and not allow
> CodeIt do to likewise.
>
> > ablation on the amount of sampled data, rather than the
> context length. How would the model perform if you reduce the number
> of samples, or even better, can you show the curve of perf vs. num of
> samples per task?
>
> There are two ways to interpret this question. First,
> if the reviewer asks about the effect of the sampled programs per task
> $n_\rho$ per meta-iteration: we chose this number based on experiments on a
> custom validation set. We did not ablate this choice here, as doing so
> would require tuning other parameters as well (learning rate, batch size
> and/or number of iterations of continual learning).
>
> Second, if the reviewer asks about performance over time, but shown on a
> different axis: we sample equally many programs per task during search,
> $n_\rho$. This means that dividing the number of sampled programs on the $x$-axis in
> Figure 3 by the number of tasks (400) will give the performance vs the number
> of sampled programs per task.
>
> >you have incorporated quite a lot of sampled programs, and
> I seriously doubt what would happen when the newly sampled programs
> are used by other methods. Also a pretrained CodeT5+ is used
>
> Likely, an approach in which a capable enough LM is trained on the programs
> sampled by CodeIt would be capable of imitating CodeIt’s performance;
> but this approach would amount to nothing more than a behavioral clone
> of CodeIt.
>
> We are unable to control for the amount of data used to train CodeT5+,
> since this information is not in the public domain (only the number of files used is reported, but not the amount of tokens in each). Our ablations show that using pre-trained weights does play an important role (Figure 3, right
> side).
>
> A recent work [Gendron 2023] reports a performance of 11.9% of GPT-4
> on the ARC eval set. While this is hard to compare directly to our result
> due to methodological differences (we do train on ARC, but see much less
> data), it is telling that we match one of the most capable generalist LLMs in performance, despite using a much smaller CodeT5+
> model.
>
> > Experiments only on ARC are not necessarily sufficient to
> show the superiority of the method. I knew of the Raven matrices that also
> stress abstract reasoning, and the RAVEN dataset has similarly structured
> data and a much simplified program structure. Would it be possible to
> show similarly improved performance on this task?
>
> Thanks for pointing out this benchmark, we were not aware of it. We chose the ARC
> benchmark because of its difficulty (as also remarked by reviewer NHnw),
> the availability of neural baselines to compare with, and the possibility
> to efficiently represent the grids. While the RAVEN tests are similar in
> spirit, choosing how to represent its tasks is an open question, meaning
> that applying our method would be a new research effort. We consider
> this beyond the scope of this work, but will consider it for future work.
>
> >Would it be possible to jointly search over the DSL space,
> maybe starting simple such as using a mutation method for the DSL
> space?
>
> We see learning or refining the DSL as an interesting area for
> future research. Previous work [Ellis 2020] proposes an approach called
> DreamCoder that can perform DSL refinement (referred to as abstrac tion), and work [Banburski 2020] applies this method to ARC; however,
> this scaled poorly to the size of most ARC problems. We therefore deemed
> it to be beyond the scope of the present work, and focused on an approach
> that scales well for a given DSL.

---

> > ### Comment · Reviewer_xY18 · 2023-11-23
> >
> > I have read the authors' response. I believe some questions are unanswered, and yet some are not, eg. the performance curve. Therefore, I keep my initial rating.

---

> ### Author Response · Authors · 2023-11-23
>
> Thank you for reading our response. Apart from the question on additional benchmark tasks and refining of the DSL, which questions do you believe remain unanswered? We will do our best to respond now and/or address these in the paper.

---

### Official Review · Reviewer_JPeY · 2023-11-01

**Soundness:** 3 good
**Presentation:** 3 good
**Contribution:** 2 fair
**Rating:** 5
**Confidence:** 4

**Summary:**

This work proposes an iterative program search method to tackle the challenging benchmark, ARC, designed to measure AI skill acquisition and generalization ability. The main idea is similar to iterative policy improvement approaches, where at each iteration,1)  a set of solutions are sampled from current policy, 2) local search is performed around the sampled solutions (e.g., program mutation), 3) policy is improved via learning. The model is finetuned from the pretrained Code-T5 model, and the entire work is implemented based on the DSL manually designed for ARC. The experiment shows that the proposed method, CodeIt, outperforms previous SOTA method by large margin. Also, authors conduct various ablation studies to analyze policy’s capacity to understand input-output grids (i.e., few-shot program inference ability), and the effect of policy update and pretrained weights.

**Strengths:**

* The proposed idea is simple and reasonable
* The empirical study clearly demonstrates the effectiveness of the proposed method
* The paper reads well (but there are few missing details. Please refer to the Questions section)
* The ablation experiments are adequately designed to analyze the effect of each component

**Weaknesses:**

* Limited contribution and novelty

 As described in the related work section, the proposed idea shares the main idea with the iterative policy improvement works: iterating between policy-guided search followed by imitation learning to improve the policy. It is indeed somewhat beneficial to the readers to show that applying existing idea to the new challenging domain works well. However, it would be more helpful to provide more intuition beyond that, given that the idea is mostly inspired by the existing work in other domains. For example, analyzing whether there is any unique challenge/benefit in applying iterative policy improvement ideas to the ARC domain compared to conventional RL domains could be an interesting contribution.

* Scalability is limited

 In the discussion section, authors mention that “..via program mutation and hindsight relabeling, thus requiring only a small amount of expert programs to start training”. Although this is much better than requiring a large number of programs as data, the program mutation is only possible and effective if the DSL is efficiently designed by domain experts and also the program mutation algorithm is carefully designed by domain experts. Also for filtering, the execution engine for DSL is required as well. Overall, these requirements limit the scalability of the proposed approach, and it is important to study how well the proposed method will perform depending on the quality/availability of these prerequisites.

* Comparison with baselines in common settings

 Although I recognize that there seems no widely accepted common experiment  setting exists, when it comes to comparing with other approaches, it would be more convincing if the proposed method and the compared methods are compared in a common setting. I do agree that the proposed DSL, grid representation, program mutation, and DSL execution engines are the important contributions of this work. However, still for evaluating the effectiveness of the proposed *learning framework* alone, including an apple-to-apple comparison result would greatly improve the paper’s significance. Also, comparing with other learning-based baselines would be great.

* Intuition behind recency-based sampling of input-output pairs

 It would be helpful to provide intuition behind the proposed design choice such as recency-based sampling of input-output pairs.

**Questions:**

* Comparison with previous iterative policy improvement methods

 It is not clear how the proposed method differs with iterative policy improvement. Sampling from policy and performing local search (e.g., program mutation in CodeIt) around the sampled action is a common approach in iterative policy improvement. The related work only explicitly compares with ExIt and ReST among iterative policy improvement methods.

* First paragraph of Section 3.1: the meaning of $n_\rho$ and $n_{\text{tasks}}$ are not defined

* In Figure 3, the meaning of x-axis is undefined and unclear. Section 3.3 mentions “across meta-iterations in Figure 3.”. Does it mean “number of sampled programs” is the same as meta-iterations?

* The agents “CodeIt: mutated+policy” and “CodeIt: policy only” are not defined

* Suggestion: population-based policy searching

As indicated by the ablation A2 experiment, policy improvement plays an important role in program searching. It would be an interesting future direction to try population-based policy searching approaches; i.e., maintaining multiple population of policy networks Q to enable more efficient exploration in program search space.

---

> ### Author Response · Authors · 2023-11-20
> **Response to reviewer JPey**
>
> We thank the reviewer for their comments. Updates to the paper related to this review are listed in ```brown```.
>
> > Limited novelty; the proposed idea shares the main idea with the iterative policy improvement works
>
> We respond to this point in the global response above, as novelty was raised by reviewer WdyM as well.
>
> > Scalability is limited: it is important to study how well the proposed method will perform depending on the quality/availability
> of [the DSL, mutation procedure, expert programs]
>
> It is true that our method relies on access to some expert programs and a DSL amenable to mutation. We investigate the effect of the mutation procedure in Figure 3, and through additional ablations in Appendix C, also see the global response. We observed that mutation is an essential ingredient for fast training - it is not impossible to train CodeIt without it, but it will take longer. For many program synthesis works, access to an interpreter is a necessary requirement, and as such we do not see this as a limitation.
>
> > Comparison with baselines in common settings
>
> We have added comparisons to additional neural baselines to the revised paper, in particular LLM-based methods that solve a variation of the ARC eval set.
>
> > Intuition behind recency-based sampling of input-output pairs
>
> We saw some evidence that as we ran CodeIt, programs would become shorter over time. This likely meant those solutions were of higher quality, and that they should therefore be sampled more often. We include quantitative proof of this phenomenon in Appendix D.3, where we show that for 81% of tasks for which CodeIt finds a solution, a shorter solution is found in a later meta-iteration. We also include qualitative examples in Appendix D.2.
>
> > Population-based policy searching
>
> Thanks for the suggestion to explore population-based search, this could indeed be a way to increase data diversity during exploration and prevent stagnation. For example, [Jung 2020] shows that population-based learning can be an effective way to improve off-policy RL performance. Given the limited rebuttal period, we decided to leave more advanced search approaches for future work.
>
>
> Other comments:
> - We added the definition of $n_\rho$ (number of sampled programs per task per meta-iteration) and $n_{tasks}$ (number of tasks) to the text.
> - The x-axis of Figure 3 shows the total number of sampled programs. As we sample a fixed number of programs per task per meta-iteration, dividing this number by $( n_\rho \times n_{tasks}) $ gives the total number of meta-iterations.
> - We added more descriptive names for the methods in Figure 3. Also see the global response for more info on the updated Figure 3.
>
> References:
> - [Jung 2020] "Population-Guided Parallel Policy Search for Reinforcement Learning", Jung et al., ICLR 2020.

---

### Official Review · Reviewer_WdyM · 2023-11-01

**Soundness:** 2 fair
**Presentation:** 2 fair
**Contribution:** 2 fair
**Rating:** 6
**Confidence:** 3

**Summary:**

The paper focuses on the task of programming by examples. The authors propose a method where a language-model-based policy network generates DSL programs given the input examples. The policy network is first pretrained on human-annotated examples-program pairs, and then iteratively trained on programs and input examples generated from the last iteration of the network. Program mutation and hindsight relabeling increase the diversity of the training data. The proposed method has achieved SOTA performance on the evaluation split of the ARC benchmark.

**Strengths:**

- The method is simple and straightforward.
- The proposed method achieves SOTA performance on the evaluation split of ARC. The ARC benchmark being tackled is known to be challenging. Most works do not evaluate the full evaluation dataset but instead focus on a simpler subset.
- Ablation studies are also conducted to demonstrate the contribution of each component in the model.

**Weaknesses:**

- Technical novelty is limited. As the related work mentions, iterative policy improvement and hindsight relabeling are not new ideas. In the regime of program synthesis, the idea sampling from the policy, filtering by execution, and then retraining is also explored in [1].
- The effectiveness of the proposed pipeline is not convincing:
    - From Table 1, we can see that the full version of CodeIt solves only 1% more programs compared with the “mutation only” baseline (40/400 vs 36/400). This naive baseline simply samples programs from the training dataset perturbed by changing one line of code. This indicates the DSL and the original reference program for the training dataset play a major role in achieving SOTA performance. The author also mentions this point in the discussion. But I believe it is important to expand on this.
    - To better illustrate the performance of CodeIt, I strongly encourage the authors to provide the following results: 1. how many solved tasks are in common for CodeIt and the mutation baseline? 2. For tasks that both methods solve, are there any differences between the solutions of each method? Will CodeIt generate a more succinct program?

Overall, I believe the technical novelty and the effectiveness of the method need more justification. I am willing to raise my score if my concern is addressed.

[1] Language Models Can Teach Themselves to Program Better.  Patrick Haluptzok, et al. ICLR 2023

**Questions:**

- It would be helpful if the authors could provide quantitative results on how well CodeIt compresses programs like the example in Figure 4. How many correct programs are shorted throughout the iteration? Are there programs that become longer?
- The task mutation procedure plays a crucial part in the procedure, I believe it’s important to expand the pseudocode to explain more details about it. There is no explanation about functions in the pseudocode, though some can be inferred from the names. How is this mutation procedure determined? How sensitive are the results w.r.t to the hyperparameter $\phi$?
- How do authors determine the number of training epochs during continual learning? Will training for more than 1 epoch for each iteration degrade the performance?

Other Comments:
- Figure 4, it’s hard to interpret the figure because the task being considered is not shown and the meaning of each function is unknown.
- why the example input and output are named S_0 and S_N?
- It is encouraged for authors to include the DSL in the paper, the original write-up of the DSL is not straightforward to find.

Typo
- The first sentence of Section 3.3: “We report performance of CodeIt after [sampling] 500,000 programs,”
- Second paragraph of Section 2.2: “We [finetune] the policy network on the resulting set, and initialize our replay buffer with it”. “finetune” should be “pre-train”.

---

> ### Author Response · Authors · 2023-11-20
> **Response to reviewer WdyM**
>
> We thank the reviewer for highlighting the simplicity of our method, as well as its performance and the exhaustiveness of our ablation studies. Updates to the paper related to this review are listed in `green`.
>
> > Technical novelty is limited. As the related work mentions, iterative policy improvement and hindsight relabeling are not new ideas. In the regime of program synthesis, the idea sampling from the policy, filtering by execution, and then retraining is also explored in Haluptzok et al. (2022)
>
> Thanks for pointing out reference [Haluptzok 2022], we have included a comparison in the related work section. The key differences between that work and ours are the way tasks are proposed and how output programs are
> used. In their work, synthetic tasks are proposed by a language model,
> and programs that solve those puzzle correctly are saved. In our work
> we use real task inputs only, but use hindsight relabeling to augment the
> outputs: we save all programs that can be executed and their outputs as
> new, synthetic tasks. Whether these techniques are complementary is an
> interesting question for future work.
> > The effectiveness of the proposed pipeline is not convincing:
>
> CodeIt (full eval) solves 48 tasks compared to 41 tasks in our
> Mutation $d_1$ baseline. While this is only a 7 task difference, this represents an increase of 17%. Further, we designed the mutation procedure
> as a strong baseline that makes effective use of the DSL, and tuned its
> hyperparameters – it is not a naive random search. Although deployment
> it is not a focus of this work, also note that running inference on a never
> seen-before task is hard for the mutation baseline: the best one can do is
> execute-and-check all found programs. On the other hand, CodeIt distills
> knowledge about the ARC tasks into model parameters, which enables
> running a forward pass on the new task. CodeIt’s better performance
> demonstrates that distilling knowledge about the tasks ultimately trumps
> brute-force search of a solution, even with a DSL explicitly designed to
> enable search. While CodeIt (full eval) significantly outperforms Ainooson
> et al. (2023) and Mirchandani et al. (2023), it also performs in line with
> GPT-4 Gendron et al. (2023). Achieving similar performance here speaks
> to the efficiency of the proposed pipeline since GPT-4 is a much larger
> model, with a substantially larger context window, and trained on a much larger amount of data.
>
> >  I strongly encourage the authors to provide the following
> results: 1. how many solved tasks are in common for CodeIt and the
> mutation baseline? 2. For tasks that both methods solve, are there any
> differences between the solutions of each method? Will CodeIt generate a
> more succinct program?
>
> We thank the reviewer for this suggestion. We
> provide an analysis in this spirit in appendix D.2 of the revised paper, and we refer to it
> in the global reply. In short: there are differences between the solutions,
> CodeIt programs tend to be shorter, and some tasks are only solved by
> one of the two methods.
>
> > provide quantitative results on how well CodeIt compresses
> programs like the example in Figure 4
>
> We carried out this analysis and we provide it Appendix D.3 of the revised paper. We observe
> that for 80% of tasks where a solution is found, CodeIt finds a shorter solution at a later point during the search.
>
> > expand the pseudocode [of the mutation procedure]
>
> We have expanded and better outlined the mutation algorithm as requested.
>
> > How do authors determine the number of training epochs during continual learning
>
> We tuned all of our hypeparameters on a custom validation set as described in Appendix B.2. On this set, we observed that training longer can lead to overfitting and is ultimately not beneficial.
>
> > Figure 4, it’s hard to interpret the figure because the task being considered is not shown and the meaning of each function is unknown
>
> We updated the figure and figure caption. We hope that the figure is easier to parse now.
>
> > why the example input and output are named $S_0$ and  $S_N$
>
> A CodeIt solution can be thought of as a sequence of $S$(tate) transformations, going from the 0th input grid to the desired output grid. This structure is effectively promoted by the design of the DSL. Since the number of transformation steps is unknown, we use a placeholder $N$ for the output grid, $S_N$
>
> > It is encouraged for authors to include the DSL in the paper, the original write-up of the DSL is not straightforward to find.
>
> We provide additional info and a link to the original write-up on the DSL in Appendix B.4. As the full DSL would effectively be larger than the current paper+appendix, we chose not to include it in our manuscript, but provide a link to Hodel's GitHub instead.
>
> References:
> - [Haluptzok 2022]: "Language models can teach themselves to program better", ICLR 2023

---

> > ### Comment · Reviewer_WdyM · 2023-11-23
> >
> > Thank authors for addressing my concern. I will raise my score to 6. It is very interesting to see that the policy is able to compress programs during the process without any regularization. I encourage the authors to include a discussion about potential explanations for this. Also, it is strongly encouraged to include the example intput/output examples of programs in Table 5 and Table 6 of the appendix as they make the table more interpretable.

---

### Author Response · Authors · 2023-11-20
**Global response (Part 1)**

## Global response (Part 1)

We thank the reviewers for their in-depth reviews and constructive comments. We group common points below, and respond to points raised by individual reviewers under their respective comments.

Updates to the paper are color-coded:
- General updates are `blue`.
- Updates for reviewer WdyM are `green`.
- Updates for reviewer JPeY are `brown`.
- Updates for reviewer xY18 are `orange`.
- Updates for reviewer NHnw are `purple`.

### Technical novelty (WdyM, JPeY)

Reviewers point out that iterative policy improvement and hindsight relabeling have been used in other settings. While true, we demonstrate a practical way to combine program mutation, policy improvement and hindsight relabeling, and apply this to a particularly challenging domain where training data is scarce and inter-task generalisation is required for success. When done correctly, we show that the resulting approach outperforms neural search approaches on ARC. We include additional baselines (including [Haluptzok 2022], LLM variations from [Mirchandani 2023, Gendron 2023]) in the related work section.

### Comparison to mutation in Figure 3

Reviewers ask about the influence of the mutation procedure on CodeIt performance, and what role its design choices play. They also point out the high performance of the mutation-only baseline. To better show the effect of mutation, we 1) ran an additional variation of the mutation baseline, 2) added a random baseline, and 3) ran an evaluation of CodeIt that includes mutated programs as candidate solutions "CodeIt: full eval". The updated results are shown in Figure 3 in the main text.

**1. Additional mutation variation:** Our original mutation algorithm repeated three steps: sample a program from the population, mutate, add the result to the population - potentially mutating previously mutated programs ("depth $\infty$"). When examining this baseline, we identified a key mistake: only programs with edit distance 1 to training data were accepted, the remainder were seen as invalid. This meant that instead of sampling 19,200 programs, we only sampled 3,038 valid ones.
Correcting this mistake did not change final baseline performance (9%).
However, we also ran a variation that only mutates the original training set ("depth 1"). This baseline proved stronger and faster, and we include "mutation $\(d_1\)$" and "mutation $\(d_\infty\)$" for transparency. We also expanded the description and pseudocode of both mutation procedures in Appendix A.2.

**2. Random search:** Second, we added a random search baseline in Figure 3 -- mainly to show that the mutation procedure is not a naive search, as Random performs worst overall. We hand-designed the mutation baseline by exploiting our knowledge of the DSL, and tuned its hyperparameters on our custom validation set.

**3. CodeIt evaluating full buffer:** Lastly, during CodeIt evaluation, we originally only considered programs sampled by the policy. Based on the analysis of programs, we noticed that CodeIt and mutation often produced distinct solutions. We therefore ran an additional evaluation of our existing CodeIt runs, where we include the 3,038 programs from the initial program mutation step as candidate solutions. We show in Figure 3, "CodeIt: full eval" that this outperforms all previous approaches. We include more details on this evaluation procedure, and additional ablations that use it, in Appendix C.

We did not run additional experiments for CodeIt here, only for the baselines. If reviewers feel that changes to the main results are too substantial, we could place the "CodeIt: full eval" curve in the Appendix only. The experiment "CodeIt: policy eval" is our old result, and all further analyses are (still) based on this method for consistency.

### Additional baseline comparisons (WdyM, JPeY, NHnw)

In Table 1, we added comparisons to other learned methods, [Mirachandai 2023] and [Gendron 2023]. The latter is a very recent work that applies GPT-4 to ARC. Both works report pass@1 performance on a different set, requiring a new column in the table. Although neither work explicitly optimizes for ARC, they use much larger models that have seen substantially more data. We add more details on their evaluation in Appendix A.4.

We also perform qualitative analysis of found programs, see the global response part 2 below.

### References:

- [Haluptzok 2022]: "Language models can teach themselves to program better", ICLR 2023
- [Mirchandani 2023]: "Large language models as general pattern machines", CoRL 2023
- [Gendron 2023]: Large language models are not strong abstract reasoners", 2023

---

> ### Author Response · Authors · 2023-11-20
> **Global response (Part 2)**
>
> ### Analyzing programs (WdyM, JPeY, xY18)
>
> Reviewers ask to see example programs, and ask whether we can show qualitative differences between programs found by CodeIt versus the mutation baseline (which also uses the DSL). In Appendix D, we provide in-depth analysis of the programs produced by CodeIt and the baseline. We provide a brief summary here.
>
> The main questions we tried to answer were:
>
> 1. Is there a quantitative difference between tasks solved by CodeIt and those solved by mutation? (Appendix D.1)
> 2. When a task is solved by both CodeIt and mutation, is there any qualitative difference between the solution programs? (Appendix D.2)
> 3. Does CodeIt learn to make programs shorter as training progresses? (Appendix D.3)
>
> The answer to all three questions is "yes".
>
> Compared to the mutation baseline, we observe that CodeIt tends to produce shorter solutions on average. We also note that CodeIt more often solves tasks with short task representations (i.e. smaller grids). Since the task representations need to be provided as context in the encoder, whose context length is limited, this is perhaps not surprising.
>
> When both methods solve a task, CodeIt more often than not produces a shorter solution, sometimes significantly so. We report one such case as an example in the following table, more in Appendix D Table 5; note that one cannot obtain the CodeIt solution from the mutation one just by removing redundant lines: the two solutions use different DSL primitives. Nevertheless, in some cases CodeIt's solution is longer than the mutation one, and for fairness we report such cases in Table 6 in the appendix as well.
>
> **Example comparison of CodeIt and mutation solutions for the same task:**
>
> We show an example CodeIt solution (left column) and a mutation solution (right column) for the same task. The CodeIt solution is much shorter than the one obtained via mutation.
>
> | CodeIt                         | Mutation \($d_1\$)                 |
> |--------------------------------|----------------------------------|
> | x1 = replace(I, EIGHT, ZERO)   | x1 = asindices(I)                |
> | x2 = compress(x1)              | x2 = fgpartition(I)              |
> | O = trim(x2)                   | x3 = rbind(greater, TWO)         |
> |                                | x4 = compose(x3, size)           |
> |                                | x5 = sfilter(x2, x4)             |
> |                                | x6 = totuple(x5)                 |
> |                                | x7 = apply(color, x6)            |
> |                                | x8 = apply(center, x6)           |
> |                                | x9 = pair(x7, x8)                |
> |                                | x10 = fill(I, ZERO, x1)          |
> |                                | x11 = paint(x10, x9)             |
> |                                | x12 = rbind(greater, ONE)        |
> |                                | x13 = compose(dedupe, totuple)   |
> |                                | x14 = chain(x12, size, x13)      |
> |                                | x15 = sfilter(x11, x14)          |
> |                                | x16 = rot90(x15)                 |
> |                                | x17 = sfilter(x16, x14)          |
> |                                | O = rot270(x17)                  |
>
>
>
> Finally, we confirm that CodeIt does compress programs as training progresses: in 80\% of cases, a solution will get shorter in subsequent meta-iterations after it's first discovered.
> We show more examples of this phenomenon in Table 7 in the Appendix.
> We do not explicitly reward for program length, but do favor shorter programs during training (sampling shorter programs more often) and evaluation (if two solutions have equal performance on demonstration examples, we take the shorter ones).
>
> **Example of program compression through meta-iterations:**
>
> Selection of longest (left) and shortest programs (right) for ARC evaluation tasks solved by CodeIt.
>
> | Long program                    | Short program                    |
> |---------------------------------|----------------------------------|
> | **Task key: e133d23d**          |                                  |
> | *Longest (Iteration 27)*        | *Shortest (Iteration 50)*        |
> | x1 = lefthalf(I)                | x1 = vmirror(I)                  |
> | x2 = righthalf(I)               | x2 = lefthalf(I)                 |
> | x3 = ofcolor(x1, ZERO)          | x3 = righthalf(I)                |
> | x4 = ofcolor(x2, ZERO)          | x4 = cellwise(x2, x3, TWO)       |
> | x5 = intersection(x3, x4)       | O = replace(x4, SIX, SIX)        |
> | x6 = astuple(THREE, THREE)      |                                  |
> | x7 = canvas(TWO, x6)            |                                  |
> | x8 = fill(x7, ZERO, x5)         |                                  |
> | O = replace(x8, SIX, TWO)       |                                  |

---

### Meta-Review · Area_Chair_7dfq · 2023-12-06

**Metareview:**

This work proposes a methodology for training neural networks to synthesize programs from input-output examples in a domain specific language. The main idea is to start with a small dataset of (problem,program) pairs, and then bootstrap your way to a large synthetic dataset using tricks like program mutation and hindsight replay, essentially constructing for itself a large augmented training set.

The primary empirical result is that, relative to other neural-network approaches, the resulting system does well on an interesting benchmark called ARC, solving 12% the evaluation problems.

The obvious strength is the authors evaluate on the harder *evaluation* ARC split, which very few papers do. Most other papers applying neural networks to ARC instead report numbers on the much easier training split. The performance level is genuinely impressive. Arguably, still inferior to purely symbolic methods: For instance, a well-known Kaggle-winning 2019 system solves more than twice as many problems from the evaluation set (https://github.com/top-quarks/ARC-solution); and the only symbolic baseline compared with, Ainooson et al., uses a different DSL, so does not serve as a good test of how much their learning method actually helps. The authors use a mammoth DSL.

One weakness, identified by some reviewers, is that the method is not particularly novel. One strength, however, is that the method is very simple and clean.

**Justification For Why Not Higher Score:**

Although there are good reasons for accepting this paper, it is hard to advocate for it simply on the basis of "benchmark busting": Its actual improvement on the raw benchmark is small, arguably its performance is not yet at the level of purely symbolic methods, and ARC is niche (although cool!). Unfortunately, there are not the kinds of interesting ideas that might otherwise justify publication.

**Justification For Why Not Lower Score:**

n/a

---

### Decision · Program_Chairs · 2024-01-16

Reject